# Synthesis and Properties of Fully Biobased Crosslinked Starch Oleate Films

**DOI:** 10.3390/polym15112467

**Published:** 2023-05-26

**Authors:** Laura Boetje, Xiaohong Lan, Jur van Dijken, Michael Polhuis, Katja Loos

**Affiliations:** 1Zernike Institute of Advanced Materials, University of Groningen, Nijenbogh 4, 9747AG Groningen, The Netherlands; l.boetje@rug.nl (L.B.); x.lan@rug.nl (X.L.); j.van.dijken@rug.nl (J.v.D.); 2Royal Avebe U.A., Zernikelaan 8, 9747AA Groningen, The Netherlands; michael.polhuis@avebe.com

**Keywords:** fatty acid starch ester, crosslinking, UV curable, heat curable

## Abstract

Starch oleate (degree of substitution = 2.2) films were cast and crosslinked in the presence of air using UV curing (UVC) or heat curing (HC). A commercial photoinitiator (CPI, Irgacure 184) and a natural photoinitiator (NPI, a mixture of biobased 3-hydroxyflavone and n-phenylglycine) were used for UVC. No initiator was used during HC. Isothermal gravimetric analyses, Fourier Transform Infrared (FTIR) measurements, and gel content measurements revealed that all three methods were effective in crosslinking, with HC being the most efficient. All methods increased the maximum strengths of film, with HC causing the largest increase (from 4.14 to 7.37 MPa). This is consistent with a higher degree of crosslinking occurring with HC. DSC analyses showed that the *T*_g_ signal flattened as film crosslink densities increased, even disappearing in the case of HC and UVC with CPI. Thermal gravimetric analyses (TGA) indicated that films cured with NPI were least affected by degradation during curing. These results suggest that cured starch oleate films could be suitable for replacing the fossil-fuel-derived plastics currently used in mulch films or packaging applications.

## 1. Introduction

Plastics produced from fossil fuel sources find use in almost every sector, including textiles, consumer products, packaging materials, building and construction, transportation, electronics, and industrial machinery. However, since these plastics are primarily derived from petroleum, their production will become increasingly curtailed once peak oil is surpassed. Moreover, plastics are not readily biodegradable and their extreme persistence is detrimental to ecosystem vitality when released into the environment. This makes it imperative to develop sustainable, biodegradable alternatives to conventional plastics [1,2].

Starch is the second most abundant biopolymer on the planet. It is a cheap and sustainable natural product that can be readily modified with different fatty acids to create new biobased thermoplastic materials as potential alternatives to petroleum-derived plastics [3,4,5,6]. Earlier studies have demonstrated that starch oleates cast from solution provide films with high transparency and good mechanical properties. Additionally, the oleate double bonds create opportunities to further increase films’ mechanical strengths and improve their thermal properties through crosslinking [7].

Auto-oxidation enables the direct crosslinking of unsaturated fatty acids. This occurs via a reaction of double bonds with available oxygen to produce reactive hydroperoxide intermediates. These intermediates subsequently react with another double bond to form a crosslink via an ether linkage [8,9,10,11,12,13]. However, auto-oxidation can impart disagreeable flavors and odors to fats and foods containing them. This is not desirable in films that will be in contact with comestibles [11]. Consequently, auto-oxidation as a means of crosslinking films for food packaging materials is preferably avoided.

One facile alternative to auto-oxidation for inducing the crosslinking of unsaturated fatty acids is to use light [11,12]. Light-driven crosslinking is relatively fast, low in cost, and highly efficient. Moreover, the process can be expedited by employing a suitable photolabile species that functions as an initiator for the crosslinking reaction. Photoinitiators are classified as Norrish Type I and II according to how they initiate crosslinking or polymerization.

Norrish Type I photoinitiators, e.g., aromatic α-hydroxyketones, undergo homolytic cleavage into two radical fragments upon irradiation with UV light. Both radicals can initiate crosslinking or polymerization [14].

Norrish Type II initiating systems, on the other hand, comprise two species. One species is the actual photoinitiator, usually an aromatic ketone. The other species is a synergistic hydrogen donor, frequently an amine. The overall mechanism of photoinitiation involves the initial excitation of the photoinitiator upon UV irradiation followed by H abstraction from the synergist. The synergist, now containing a radical center, has an enhanced reactivity towards double bonds compared to the excited photoinitiator. This makes the synergist more effective at initiating crosslinking or polymerization compared to the photoinitiator [15,16].

Crosslinking can also be induced by heat curing (HC), a simple process that only requires the application of heat to cast films. This is contrary to UV curing (UVC), which requires the addition of a photoinitiator followed by thorough mixing before casting films that are to be cured. For this reason, HC is the preferred method of polymer curing in industry. One example of HC is the crosslinking of oleic-acid-based polymers by thermal means [17]. Another advantage of HC over UVC is that it is not limited by sample thickness. Furthermore, HC crosslinking occurs at temperatures above the materials’ *T*_g_. This permits the relaxation of the polymeric chains after crosslinking due to mobility induced by heat. UVC, on the other hand, is performed below the *T*_g_. This impacts oleate chain mobility, limiting the crosslinking efficiency [18]. Thus, HC can be expected to produce films possessing better mechanical properties compared to UVC.

After successfully preparing non-crosslinked starch oleate films with good mechanical and thermal properties, we have now investigated the crosslinking of these films using both HC and UVC. For the UVC experiments, both Type I and Type II photoinitiating systems were investigated. The Type I photoinitiator is a commercial photoinitiator (CPI, Irgacure 184). The Type II system is a natural photoinitiator (NPI, a mixture of 3-hydroxyflavone as the main photoinitiator and n-phenylglycine as the synergistic H-donor) which has successfully been used for the free radical polymerization of acrylates [19]. Our choice of 3-hydroxyflavone and n-phenylglycine was prompted by them both being natural products extracted from plants, potentially enabling the formation of fully biobased films, in contrast to non-biobased Irgacure 184.

The mechanism of radical formation for both photoinitiator systems is shown in Figure 1.

## 2. Materials and Methods

### 2.1. Material

Pregelatinized (cold water swellable) native potato starch (Paselli WA4) was supplied by Avebe and dried at 110 °C overnight prior to use. Oleic acid (90%), 1,1′-carbonyldiimidazole (CDI, ≥97%), chloroform-d3 (CDCl_3_, 99.8%), n-phenylglycine (NPG, 97%), and 3-hydroxyflavone (3HF, ≥98%) were purchased from Sigma-Aldrich (St. Louis, MO, USA). Irgacure 184 (>90%) was purchased from BASF(Ludwigshafen, Germany). All organic solvents used for syntheses have a minimum grade of ≥99.7%. All chemicals were used as bought unless stated otherwise.

### 2.2. Methods

#### 2.2.1. Synthesis of Starch Oleate

Starch oleate films were prepared according to a known method [20]. Predried potato starch was added to a round-bottomed flask and dissolved in DMSO at 80 °C. Oleic acid was added to a second round-bottomed flask and dissolved in DMSO followed by activation with 1,1′-carbonyldiimidazole (CDI) for 30 min at 60 °C. This solution was transferred to the starch solution and the mixture was stirred overnight at 80 °C to obtain an insoluble gel. After decantation of DMSO, the gel was dissolved in CHCl_3_ and precipitated in ethanol (3x). The final precipitate was collected and dried in a vacuum oven at 40 °C overnight to obtain starch oleate (degree of substitution = 2.2 ± 0.1).

^1^H-NMR (400 MHz, CDCl_3_, δ, ppm): 5.39, (1 H, br, -C3-O**H**, starch), 5.31 (2 H, br, -C**H**=C**H**-, oleate double bond), 5.27 (1 H, br, -C2H-O**H**, starch), 4.75 (1 H, br, -C1**H**-, starch), 4.43 (1 H, br, -C6H_2_-O**H**, starch), 4.08 (2 H, br, -C6**H**_2_-, starch), 3.97 (1 H, br, -C3**H**-), 3.87 (1 H, br, -C5**H**-), 3.68 (1 H, br, -C2**H**-, starch), 3.56 (1 H, br, -C4**H**-, starch), 2.33 (2 H, br, -O-C**H**_2_-, oleate), 1.99 (4 H, br, -C**H**_2_-CH=CH-C**H**_2_-, oleate), 1.58 (6 H, br, -O-CH_2_-C**H**_2_-, -C**H**_2_-CH_2_-CH=CH-CH_2_-C**H**_2_-, oleate), 1.25 (14 H, br, -CO-CH_2_-CH_2_-C**H**_2_-CH, oleate), 0.86 (3H, br, CH_3,_ end group of fatty acid).

#### 2.2.2. HC Films

Films were cast by dissolving 1.5 g of starch oleate in 50 mL of CHCl_3_ followed by sonication for 30 min in a water bath at room temperature. The degassed solution was poured into a square Teflon tray (10 × 10 cm) on a 40 °C hotplate, producing a transparent film once all CHCl_3_ had evaporated [18]. The film was then cut into dog-bone-shaped halters and placed in a 150 °C air-forced oven for either one, two, three, or four hours to crosslink in air.

#### 2.2.3. UVC Films

Films were prepared as above except that a photoinitiator (1.5 wt%) was added to the casting solution and casting was performed in the dark. For films cast with the CPI, 22.5 mg (4.60 mmol) of Irgacure 184 was added. For the NPI, a mixture of n-phenylglycine (15.5 mg, 2.35 mmol) and 3-hydroxyflavone (7.6 mg, 1.81 mmol) was added [19]. Films were dried in the dark followed by irradiation under a UV lamp (two light tubes of 8 W) at 366 nm for 100 s, one hour, or two hours, corresponding to an energy supply of 1.6 KJ, 57.6 KJ, and 115.2 KJ, respectively. The lamp was placed directly on a 1 cm high Teflon mold.

### 2.3. Characterization

#### 2.3.1. ^1^H-NMR Spectroscopy

^1^H-NMR spectra were recorded in ppm relative to the solvent peak (chloroform-d_6_, 7.14 ppm) on a Varian 400 MHz spectrometer. The degree of substitution (DS) of starch hydroxyl groups was calculated using Equation (1) [18].
3DS/(2DS − (3 − DS)) = 3/X, (1)

Signal integration was normalized to the 3 protons of the terminal -CH_3_ group of the fatty acid chain (0.8 ppm). X is the integrated value of all signals between 4.7 and 6.0 ppm.

#### 2.3.2. Fourier Transform Infrared Spectroscopy (FTIR)

FTIR spectra were measured using the attenuated total reflectance (ATR) mode on a vertex 70 Bruker spectrometer. A resolution of 16 cm^−1^ and a scan time of 32 s were used for both sample and background scans. The background was corrected and the spectra normalized to the peak at 1030 cm^−1^ with OPUS software.

#### 2.3.3. Film Gel Contents

Film gel contents were measured according to a previously published method [21]. Films were weighed (initial weight = *m*_0_) before immersion in a vial with CHCl_3_ for 48 h. CHCl_3_ was decanted from the vial and the swollen fraction that was left behind was dried overnight under vacuum at 40 °C. The films were weighed again (final weight = *m_d_*) and gel contents were calculated using Equation (2).
Gel content (%) = *m_d_*/*m*_0_ ∙ 100% (2)

#### 2.3.4. Tensile Testing of Films

The films were cut into dog-bone-shaped halters before testing on an Instron 5565 tensile tester using a 100 N load cell and a pulling speed of 2 mm∙sec^−1^. Care was taken to ensure that the middle part of the halter was always smooth to avoid the samples breaking along small cracks that were already present beforehand. In addition, every measurement was run such that the sample broke between 1 and 3 min when performing the test to ensure data quality. The average from five samples was taken, including the standard deviation.

#### 2.3.5. Differential Scanning Calorimetry (DSC)

DSC measurements were performed on a TA-Instruments Q1000. The samples were placed in a non-hermetically sealed aluminum pan and analyzed using heat–cool–heat cycles ranging from −20 °C to 200 °C at a rate of 10 °C∙min^−1^. The glass transition temperature (*T*_g_) was determined using TRIOS software (TA instruments).

#### 2.3.6. Thermogravimetric Analyses (TGA)

Degradation temperatures were determined by TGA using a TA-instrumental 5500. Samples were kept under a nitrogen atmosphere while being heated from 25–700 °C at a rate of 10 °C∙min^−1^. The degradation temperature was determined as the temperature at which the rate of weight loss was maximum.

Isothermal gravimetric analysis was used to provide more insight into film crosslink densities. The samples were heated to 150 °C at a rate of 10 °C∙min^−1^ and this temperature was maintained for three hours. These measurements were performed under air instead of a nitrogen atmosphere.

All data were analyzed with TRIOS software (TA instruments).

#### 2.3.7. Contact Angle

Surface wettability was established by the contact angle of deionized water (2 μL) on the surface of the films in air. The angle was calculated with a contact angle system VCA-2500XE (AST). The average from five samples was taken, including the standard deviation.

#### 2.3.8. Statistical Analyses

Means and standard deviations were calculated for the tensile test and gel fraction measurements. The data were analyzed using one-way analysis of variance (ANOVA) with Tukey’s test in SPSS^®^ statistics software (version 26, IBM, New York, NY, USA). Data are considered to be significantly different when *p* < 0.05.

## 3. Results and Discussion

### 3.1. Starch Oleate Analysis

Our previous study outlining the synthesis of starch oleate films forms the basis of the current research [3]. First, starch oleate was synthesized with CDI via Steglich esterification and its ^1^H-NMR spectrum is shown in Figure 1. A DS of 2.2 was calculated using Equation (1) in Section 2.3.1. All peaks corresponding to the oleate chain and the starch anhydroglucose units (AGU) were assigned. After esterification, every starch AGU contains approximately 8 protons, while the oleate group contains 73 protons. Therefore, the starch signals are much lower in intensity than the signals of the oleate chains. As this material is not crosslinked, the double bond at approximately 5.3 ppm is still clearly present.

### 3.2. Film Appearance

Figure 2a shows a film sample obtained by UVC, initiated with the CPI. This sample is transparent and colorless, but has a typical smell from the photoinitiator. The film cured with the NPI is transparent but has a yellow color due to the amine group of n-phenylglycine (Figure 2b) [16,19]. The HC film is transparent but has a yellow color, the intensity of which increases with curing time (Figure 2c). This yellowing is likely caused by the formation of carbonyl groups that arise from side reactions occurring during auto-oxidation [22,23].

### 3.3. Verification of Film Crosslinking

The crosslinking of starch oleate in all cases resulted in products no longer soluble in organic solvents. This prohibited the measurement of any remaining double bonds by ^1^H-NMR. Therefore, crosslinking was verified by several other techniques.

Figure 3a displays the complete FTIR spectra of the non-crosslinked starch oleate film and all cured films. The uppermost spectrum corresponds to non-crosslinked starch oleate and shows a band at 3010 cm^−1^ arising from the C-H stretch of the unsaturated bond [17,24]. This signal should disappear when the oleate chains crosslink [17]. Since this band is rather small, Figure 3b–d focus specifically on this band for CPI, NPI, and HC, respectively.

The peak at 3010 cm^−1^ decreases in intensity after curing for 100 s for both UVC methods. This means that the number of unsaturated double bonds decreases, consistent with crosslinking of the material taking place [25]. However, extending the irradiation time does not result in a further reduction in the peak at 3010 cm^−1^. By contrast, this peak has fully disappeared after two hours of HC, suggesting complete crosslinking of all double bonds. These results suggest that crosslinking via UVC is restricted by the limited penetration of UV radiation into the material. Consequently, double bonds in the bulk material do not participate in crosslinking. With HC, on the other hand, heat can be more evenly distributed, thereby enabling the participation of all double bonds in crosslinking.

Crosslinking by HC occurs via auto-oxidation and results in the formation of ether bonds. Conversely, UVC is expected to crosslink the oleate chains directly, although the formation of some ether linkages might still occur. This is attributed to the presence of oxygen both at the film surfaces and within bulk material due to the incorporation of air into the casting solutions during mixing and pouring. In all cases, the presence of formed ether linkages is expected to generate a signal at approximately 1200–1000 cm^−1^ in the FTIR spectra. However, this signal will coincide with the strong absorption peaks of starch at 1018 and 1156 cm^−1^. This, in turn, will make it difficult to ascertain with any degree of certainty the emergence of ether bonds [17,26].

Although heat or UV light is needed to crosslink the material, prolonged exposure to either of them can lead to degradation of the films. The band at 1730 cm^−1^ corresponding to the carbonyl of the oleate ester group broadens slightly with increased curing time for all curing strategies. This broadening is consistent with the formation of side products, e.g., unsaturated aldehydes, ketones, and esters, as a result of chain scissioning [24,27]. In all curing strategies, the hydroxyl absorption band at 3500 cm^−1^ increases slightly in intensity with increasing curing time, and in particular for HC. This can be attributed to a breaking of ester bonds causing the number of hydroxyl groups to increase [18]. In addition, hydroperoxyl intermediates are formed during auto-oxidation. If these intermediates fail to react with other oleate chains, an increase in the intensity of the hydroxy signal will occur [28].

Gel content measurements offer an additional approach for evaluating the extent of crosslinking. Starch oleate films are fully soluble in chloroform before curing, meaning the gel content is 0%. After curing, none of the films were able to fully dissolve. Table 1 lists the corresponding gel contents for films prepared using the different curing methods.

The film crosslinked with CPI for 100 s has a gel content above 97%, significantly higher than that of the NPI film. Longer irradiation times do not result in significant changes in gel contents in both cases. This is commensurate with the corresponding FTIR data likewise showing no significant changes in double bond signal intensities for longer irradiation times.

The lower gel contents of NPI films compared to CPI can plausibly be attributed to the former being a Norrish Type II photoinitiator, for which radical formation is a bimolecular process. Thus, radical formation for NPI requires that the initially excited 2-hydroxyflavone must subsequently encounter and abstract a H-atom from 2-phenylglycinse before initiation of crosslinking can occur. This constraint does not apply for CPI, for which a unimolecular process involving internal scission to produce two separate radical species applies. In addition, more CPI was used on a molar basis compared to NPI. This means that CPI inherently produces more radicals per unit time compared to NPI, hence a lower rate of gel formation can be expected for NPI compared to CPI. Furthermore, as crosslinking proceeds, the rates of initiation will decrease as the diffusion of radical species becomes increasingly hindered. This limitation will be more severe when a bimolecular mechanism is operative (NPI) compared to a unimolecular mechanism (CPI) [14,29].

Films formed under UV irradiation in the absence of a photoinitiator also possess a gel fraction. This is expected because the auto-oxidation of unsaturated fatty acids does occur naturally and is furthermore facilitated by UV irradiation [13]. However, the gel fractions are significantly lower compared to films irradiated in the presence of a photoinitiator. This indicates that the formation of radicals does expedite film curing.

After one hour of curing, almost the full weight of the HC film is in the gel form. Further curing leads to only slight increases in gel contents. This is again consistent with the FTIR data showing a reduction in the number of unsaturated bonds after one hour of HC, and their disappearance after further curing.

While high gel contents are attained after 100 s using UVC, the corresponding FTIR spectra clearly show the presence of unreacted double bonds. This can be explained by the fact that a single amylose or amylopectin molecule may contain multiple oleate chains that can participate in crosslinking. However, only a limited number of oleate chains need to crosslink in order for the product to become insoluble, thus the gel content will increase much faster than the actual crosslink density. Therefore, a third analytical technique was chosen to provide more insight into the degree to which the oleate chains participate in crosslinking.

Oleic acid is known to auto-oxidize and take up oxygen from the air when heated, increasing its weight slightly [30]. Since this could also occur with the starch oleate chains, the films prepared by HC and UVC were subjected to isothermal gravimetric analysis, involving heating the films at a constant temperature while continuously being weighed.

Figure 4a shows the results of the films containing the CPI, and, contrary to the non-crosslinked film, no increase in weight was measured for the film cured for 100 s. This suggests full participation of most, if not all, of the oleate chains at or near the surface of the film. After one hour of UVC, a rapid loss of weight was observed. This is ascribed to degradation of the film due to prolonged exposure to UV light. During degradation, species such as hexanal, nonanal, 9-oxononanoic acid, azelaic acid, and nonanoic acid can be formed. Several of these compounds have boiling points below the temperature at which the isothermal gravimetric analysis is performed. These species are likely to evaporate during the course of the measurements [11,29,31], giving rise to a loss of film weight.

The film containing NPI that was irradiated for 100 s showed an increase in weight (Figure 4b). This suggests that some oleate double bonds were unaffected by the initial irradiation step and thus reacted with oxygen. No weight gain was observed for films subjected to longer irradiation times, although a shoulder was still visible after one and two hours of UVC. This shoulder indicates ongoing oxygen absorption, but to a lower extent. These results show that the number of oleate chains participating in crosslinking increased with increasing irradiation time, although full curing was not reached after two hours of UVC, consistent with the gel content experiments and FTIR data.

Thus, isothermal gravimetry confirms that the curing of films containing the CPI is faster than when NPI is used. This is because more radicals that initiate curing can be, and are indeed, generated with CPI compared to NPI. However, degradation of the films is, therefore, also faster when CPI is used compared to NPI.

As a control, non-crosslinked starch oleate was exposed to UV light for the same time intervals used to cure the CPI and NPI films, but without added photoinitiator. A small increase in weight was found for all intervals, indicating uptake of oxygen (Figure 4c) and thus the presence of unsaturated double bonds. This is in agreement with the gel content measurements and again shows that the addition of a photoinitiator accelerates crosslinking. Similar to the other UVC films, degradation increased with irradiation time, but the effect was lower since no photoinitiator was present that could create radicals.

Although none of the HC films showed an increase in weight, the films after one and two hours of curing do show a shoulder in the isothermal gravimetric analysis spectra, indicating minimal uptake of oxygen (Figure 4d). After three and four hours of curing, this shoulder is fully absent. This is commensurate with the full participation of the oleate double bonds in crosslinking and congruent with FTIR and gel content results. The collective analytical data also show that curing films via heat is the most effective method for crosslinking starch oleate films.

Based on the above, it can be concluded that HC results in films with the highest crosslink density, followed by UVC with the CPI and then UVC with the NPI. Using a different curing strategy, such as electron beam irradiation instead of UV light, might positively influence the crosslink density and therefore be interesting for future research. However, it is also important to consider the potential drawbacks of this technique, such as its tendency to cleave the glycosidic bond in starch and thus negatively impact film properties [32,33].

### 3.4. Mechanical Properties

The mechanical properties of the crosslinked films were evaluated using a tensile tester. Table 2 indicates that maximum elongation is not significantly different between most samples. This is seemingly in contrast with the crosslinking of materials generally leading to a reduction in maximum elongation [34,35,36]. However, the large starch oleate molecules are likely to have had numerous physical entanglements before crosslinking. In this way, additional crosslinking will have a smaller effect on the maximum elongation because the ability of the molecules to slide alongside each other was already restricted [37]. The observed general lack of a significant difference between maximum elongation for most of the films also implies that any observed differences in the Young’s moduli are mainly caused by changes in maximum strengths.

The films crosslinked with both CPI and NPI show a slight decrease in maximum strengths when irradiated by UV light for 100 s, although this decrease is only significant for NPI. Possibly, the irradiation time was too short, such that the initiator functions more like a plasticizer [38] despite some crosslinking having occurred. No significant differences were found between the Young’s moduli of the non-crosslinked films and the UVC films after 100 s of UV irradiation, which were significantly different among themselves. This is because Young’ modulus depends on both stress and strain, the latter being the ratio between film deformation, or in this case maximum length upon elongation, and the original length of the film. As the strain of the film with NPI irradiated for 100 s increased slightly, its Young’s modulus did not change significantly compared to the non-crosslinked films. However, the NPI film is significantly different from the film with CPI irradiated for 100 s, in which the maximum elongation decreased slightly.

After one hour of UVC, the maximum strength and Young’s modulus of both CPI- and NPI-containing films increased, becoming significantly larger compared to non-crosslinked films. This is congruent with an increase in crosslinking density generally increasing the maximum strength and Young’s modulus [34,35,36].

After two hours of curing, maximum strengths and Young’s moduli show a decrease, with the highest decrease being observed for the CPI film. This is ascribed to the degradation of the material by UV light, as already mentioned while discussing isothermal gravimetric analytical data.

The HC films possess significantly higher strengths and Young’s moduli than the UVC films, again consistent with a higher extent of crosslinking of the material by HC. The results are also in agreement with isothermal gravimetric analyses, and FTIR and gel fraction measurements, and they demonstrate that curing above the *T*_g_ has a greater impact on the crosslink density. This results in a more significant improvement in mechanical properties compared to curing strategies such as UVC, where the mobility of the oleate chains is limited. The Young’s modulus and maximum strength both reached a maximum after three hours of HC. They remained essentially unchanged after an additional hour of curing (4 h), although the maximum elongation was significantly lower compared to the non-crosslinked material. This is fully consistent with a very high level of crosslinking leading to a reduction in maximum elongation. The results are also in agreement with the FTIR data showing the complete absence of double bonds in the film after 4 h of thermal curing, together with the highest measured gel content being found.

### 3.5. Thermal Properties

The influence of the different curing strategies on the films’ thermal properties was studied using DSC. Whereas native starch has no visible *T*_g_, esterification with fatty acids can be expected to produce a thermoplastic material [18,39]. It can also be anticipated that the *T*_g_ can disappear upon crosslinking, since crosslinked materials are generally thermosets [24]. This is indeed observed in the corresponding DSC spectra for the films prepared using CPI and HC, shown in Figure 5a,c, respectively.

When CPI is used, the *T*_g_ increases and becomes less pronounced after 100 s of UV irradiation. After two hours of UVC, it is no longer visible due to an increased crosslinking density of the material.

The films cast with NPI (Figure 5b) show a visible *T*_g_ for all curing times. The *T*_g_ is still visible because the crosslink density is lower compared to films cast with the CPI, as already shown by the gel content measurements and isothermal gravimetric analytical data. The *T*_g_ decreased slightly compared to the non-crosslinked starch oleate film, possibly due to the NPI acting as a plasticizer at low crosslink densities [38].

For the HC films, the *T*_g_ increased after one hour of curing time, but disappeared with longer curing times. Since isothermal gravimetric analyses, FTIR spectra, and gel content measurements show that the HC was the most effective method to crosslink the material, it is also expected to have the strongest influence on the *T*_g_.

Besides *T*_g_, film degradation temperatures (*T*_d_) were also measured using thermal gravimetric analysis (TGA). Contrary to isothermal gravimetric analyses, where the temperature is kept constant over time, the temperature was raised while measuring the weight of the sample.

Figure 6a shows that *T*_d_ of the non-crosslinked starch oleate film is slightly higher, by 5–10 °C, compared to the CPI films, irrespective of the irradiation time. Although crosslinking did not increase *T*_d_ further, this might not be a problem since common plastics in the packaging industry, e.g., polypropylene, have a comparable *T*_d_. Thus, an increase in *T*_d_ is therefore not essential for application purposes [40].

The TGA results also show a loss of 5–10 wt% at approximately 200 °C. The presence of the CPI can accelerate side reactions, e.g., chain scissioning, during UVC. Moreover, many side products are formed, e.g., hexanal, which has a boiling point of 129.6 °C [29,41].

The thermogram of the NPI film (Figure 6b) is similar to that of the non-crosslinked starch oleate film and the irradiation time again seems to have little effect on *T*_d_. The NPI-initiated films are less affected by side reactions, e.g., chain scissioning compared to films crosslinked with CPI, for the various reasons that have already been postulated.

Figure 6c shows a decrease in *T*_d_ after one hour of HC. A further, but slight, decrease is observed after 2 h of curing, while even longer curing times have hardly any additional effect. Contrary to the CPI, where degradation is mainly ascribed to chain scissioning, degradation of the HC films is primarily caused by ester cleavage. Such an earlier onset of degradation has also been observed in a polyhydroxy linseed oil crosslinked system, for which the *T*_d_ decreased slightly upon exposure to heat [18]. At approximately 380 °C, a second degradation stage was visible. This stage is assigned to the crosslinking of the oleate chains and was most pronounced for samples subjected to HC, which agrees with their higher crosslinking density [42].

### 3.6. Hydrophobicity

Contact angle measurements were used to assess film hydrophobicities. Figure 7 shows that native starch has a contact angle of <90° and is thus hydrophilic. Esterification of native starch with oleic acid increases the contact angle to >90°, thus starch oleate is hydrophobic.

CPI films are hydrophobic after one and two hours of UVC, while NPI films are hydrophobic after 100 s and two hours of UVC. CPI and NPI films exposed to 100 s and 1 h of UV light, respectively, have standard deviations such that they cannot be unambiguously assigned as hydrophobic or hydrophilic.

After one hour of HC, the film is still clearly hydrophobic, but the contact angle decreases with further curing until it becomes hydrophobic after three hours of HC. After four hours of curing, the contact angle is approximately 90°, but the standard deviation includes both the hydrophilic and hydrophobic regions. This decrease in contact angle with longer curing times can conceivably be explained by a cleavage of ester bonds during HC leading to an increase in the number of hydroxyl groups with a concomitant decrease in hydrophobicity. This is supported by the corresponding TGA and FTIR results.

## 4. Conclusions

Three different curing strategies were used to crosslink starch oleate films (DS = 2.2) in air. Two of the strategies utilized UV curing (UVC), one of which employed a commercial photoinitiator (CPI) and the other a natural photoinitiator (NPI). The third strategy made use of heat curing (HC).

Crosslinking was verified by gel content measurements, isothermal gravimetric analyses, and FTIR spectra, all of which showed that HC produces films with the highest degree of crosslinking, followed by CPI and lastly NPI. CPI films exhibited more degradation than NPI films. These differences between NPI and CPI films can be attributed to higher molar amounts of the latter being used, together with the fact that radical formation in CPI is a unimolecular process whereas for NPI a bimolecular process applies.

Tensile testing revealed a negligible effect of crosslinking on the maximum elongation of the films. Crosslinking did, however, increase film maximum strengths and therefore also the Young’s moduli. The maximum strength increased from 4.14 MPa for the non-crosslinked starch oleate film to maximum values of 4.71, 4.96, and 7.37 Mpa for films cured with CPI, NPI, and HC, respectively.

DSC data showed a flattening of *T*_g_ signals for all crosslinking strategies. In the case of HC and CPI, the *T*_g_ even disappeared with longer curing times. Films crosslinked with NPI continue to exhibit a *T*_g_ because they have a lower crosslinking density.

TGA measurements revealed a similar *T*_d_ when weight loss was at its maximum for UVC films. *T*_d_ slightly decreased for HC films due to the cleavage of ester bonds.

Contact angle measurements showed that native starch esterified with oleate is hydrophobic. UVC had little effect on the contact angle. HC was associated with decreasing contact angles due to film degradation during curing.

HC was thus the most successful strategy for crosslinking starch oleate as it had the greatest effect on the film’s mechanical and thermal properties. This is attributed to HC enabling a more complete participation of oleate double bonds in crosslinking. This is in contrast to UVC, for which only double bonds at or near the film surface participate due to a limited penetration of UV radiation into the samples.

This work shows that further modification of SO films via crosslinking can enhance material mechanical properties with retention of *T*_g_, potentially enabling the production of fully biobased films for use in mulch films or packaging materials.

## Data Availability

Data will be made available on request.

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
