# Peer review of "Synthesis and Properties of Fully Biobased Crosslinked Starch Oleate Films"

_polymers, 2023, doi:10.3390/polym15112467_

Round 1

Reviewer 1 Report

The manuscript needs thorough English editing.

Line 120-124: This piece of information required further explanation.

Line 162: Double check the subscript letter used to indicate final weight in Equation 2.

Line 207: Define SO in the caption of Figure 1.

Line 217-220: Figure 2 shows the samples prepared by a dumbbell shape. As noticed, there are a lot of fissures at the edge of the sample, which may lead to earlier failure when the samples were subjected to mechanical testing. This may lead to poor data quality.

Line 361-452: The Mechanical properties and thermal properties data is better discussed with moisture content of each films understudied. Unfortunately, moisture content data is not available.

Line 373-374: Statistical data in Table 2 is questionable. Double check and please revised all statistical analysis done to this work.

Line 455: Section 3.6 was not well analyzed. Why was it the hydrophobicity results fluctuating?

Line 459-461: The statement was not supported by any statistical data.

The manuscript needs to be submitted for thorough English editing.

Reviewer 2 Report

Manuscript Number: Polymers-2279961

The manuscript written by Boetji et al. titled “Synthesis and properties of fully biobased crosslinked starch 2 oleate films” provided a comparative study of the effect of various curing techniques of starch oleate films. After going through the manuscript, I want to suggest the following comments and clarify a few concerns which will further improve the article and make it suitable for Polymers.

1.      In the introduction, line 78-80, “Additionally, HC crosslinking takes place at temperatures above the materials’ Tg. UVC, on the other hand, is affected below the Tg, impacting oleate chain mobility and thus limiting the crosslinking efficiency”. UVC curing is fast and often results in crosslinking stress development compared to HC where the polymeric chains can relax after crosslinking due to mobility induced by heat.

2.      In 2.2.2, line 131, please indicate which type of oven was used. Was it air forced?

3.      What was the thickness of the HC and UVC films?

4.      Will it be possible to report the total energy provided during UV curing for different time intervals?

5.      Was the method used to measure the gel content was a modified ASTM method, if so please report. Also please correct equation 2 (m4 to md).

6.      The pictures could be taken using a white background and some color printing to show the transparency of the films and the color development during surfing.

7.      Line 240, Please change the word 'interior region' to the bulk material. Line 248, Remove the period after cm-1.

8.      The autooxidation of plant oil-based polymers has been extensively studied by Webster et al. The author could include the references in the introduction (https://doi.org/10.1016/j.porgcoat.2018.09.033, https://doi.org/10.1016/j.porgcoat.2022.107252) in line 45-48.

9.      Were the spectra’s in Figure 3 normalized? Please indicate.

10.  Line 263-265, the author could also mention that an increase in hydroxyl peak might also be contributed due to the formation of hydroperoxyl intermediates during autoxidation. Please look at the reference by Kalita et al., https://doi.org/10.1016/j.porgcoat.2022.106996.

11.  Line 248, please correct the word ‘adical’ to radical.

12.  Any reason for the lower gel content of film cured by UV NPI reported in Table 1. Also, it would be good for the readers to provide the DMA results of selected cured films.

13.  Please check for other corrections.

Good, and easily understandable.

Reviewer 3 Report

In this manuscript, the author reported the synthesis and properties of fully biobased cross-linked starch oleate films. Some issues were listed as follow:

(1) Line 10, what is DS? Which should be define for the first appear.

(2) Figure 1, the analysis of the NMR spectrum is wrong, most of the materials is starch, why does the peak of starch is almost invisible?

(3) The main peak of vibration in Figure 3 should be marked.

(4) What does the superscripts mean in table 1 and 2?

(5) In table 2, why does the film strength decrease for the UV CPI?

(6) Figure 5, generally the Tg will increase with the cross-linking of UV, why does the Tg deceased after UV curing? As well as the thermal stability (Figure 6, TG).

(7) Ref.3 for the author should be deleted, as it wasnt published, which is not meet the standard for References.

(8) As we all know, the penetration depth of UV is limited, and expensive photoinitiators is need, leading to the low cross-linking density, a more efficient curing technology of electron beam radiation is recommend.

Moderate editing of English language

Round 2

Reviewer 1 Report

Authors are advised to revise the way to assign superscript letters to each mean value after the Tukey's test. Normally, the mean values are arranged from highest to lowest value. And the highest value will receive the letter "a", the next value will receive the letter "b", this will go on. When mean values of no significant different will share the same letter. The current practice by authors are not a common practice and looks messy.

Author Response

Thank you for your comment. We have arranged the values in Table 1 based on the means of the gel content, sorting them from highest to lowest. In addition, we have changed the superscript such that the highest values started with “a” and the next “b” etc. as you suggested.

For Table 2 we arranged the means from highest to lowest values based on the Young’s modulus. Additionally, we changed all the superscripts in this table such that for every column the means with the highest value started with “a”, similar as in Table 1.

Reviewer 3 Report

All of the issues mentioned were resolved in detail

 Moderate editing of English language

Author Response

In the previous round of corrections / changes we did address the request of reviewer #1 to improve the quality of English. This was, apparently, to his or her satisfaction as s/he has subsequently ticked the comment box corresponding to the quality of English as being “Fine / No issues detected”.

We have nevertheless attempted a second revision of the English as per your request. We would like to mention, however, that such an undertaking is complicated by the Tracked Changes from the previously revised manuscript still being active, together with a very short turnaround of 2 days in which to re-submit the manuscript with the additional requested changes. This combination certainly curtails our ability to complete this task in a satisfactory manner.

Should our attempts at further improving the quality of English remain inadequate according to your assessment it would be most helpful if the current changes are first accepted (thus Tracked Changes are no longer present) before any further attempts might be made. In addition, it would help M. Polhuis (a native English speaker) if you can explicitly state what you (still) find objectionable, so that it can be specifically addressed to your satisfaction.